# Low-Energy Electron Scattering from *c*-C₄F₈

Dhanoj Gupta [1],*, Heechol Choi [2], Deuk-Chul Kwon [2], He Su [3], Mi-Young Song [2], Jung-Sik Yoon [2]
and Jonathan Tennyson [3]

[1]   Department of Physics, School of Advanced Sciences, Vellore Institute of Technology,
     Vellore 632014, Tamil Nadu, India
[2]   Institute of Plasma Technology, Korea Institute of Fusion Energy, 37 Dongjangsan-ro,
     Gunsan 54004, Jeollabuk-do, Korea; hcchoi@kfe.re.kr (H.C.); dckwon@kfe.re.kr (D.-C.K.);
     mysong@kfe.re.kr (M.-Y.S.); jsyoon@nfri.re.kr (J.-S.Y.)
[3]   Department of Physics and Astronomy, University College London, Gower St., London WCIE 6BT, UK;
     he.su@ucl.ac.uk (H.S.); j.tennyson@ucl.ac.uk (J.T.)
*   Correspondence: dhanojsanjay@gmail.com

**Abstract:** Electron collision cross-sections of *c*-C₄F₈ were investigated at low energies by using the R-matrix method. The static exchange (SE), static exchange with polarization (SEP), and close-coupling (CC) models of the R-matrix method were used for the calculation of the scattering cross-section. The shape resonance was detected with all the models at around 3~4 eV, and a Feshbach resonance was detected with the SEP model at 7.73 eV, in good agreement with the previous theoretical calculation. The resonance detected was also associated with the experimental dissociative electron attachment of *c*-C₄F₈, which displayed the resonances at the same energy range. The cross-sections calculated are important for plasma modeling and applications.

**Keywords:** electron scattering; R-matrix; elastic cross sections; plasma applications

## 1. Introduction

Electron-scattering studies with atoms, molecules, and ions are of great importance to understanding and modeling the low-temperature plasmas (LTPs), which play a crucial role in technological advancements [1]. *c*-C₄F₈ is widely used in thin-film etching processes, such as Si, SiO₂, HfO₂, Si₃N₄, and SiO₂-Si₃N₄-SiO₂ stacks [2–6]. It is diluted with various gases rather than a pure *c*-C₄F₈ molecule or is often used with SF₆ in a multiple-step deep-Si etching process [2]. However, since various ions and radicals are generated in *c*-C₄F₈ mixed plasmas, it is challenging to understand *c*-C₄F₈ plasmas. Therefore, many research groups have conducted research to analyze *c*-C₄F₈ plasmas and optimize the process. Li et al. analyzed experimentally the effect of mixing additional gases to *c*-C₄F₈ inductively coupled plasmas (ICP) on the oxide etch rate [3]. Experiments were conducted by mixing Ar, He, and Ne in *c*-C₄F₈ plasmas, and the results showed that the highest ion current density was obtained when Ar was mixed. Moreover, the etch rate of SiO₂ depends on the type of noble gas added to *c*-C₄F₈. Hua et al. investigated the effect of N₂ dilution on the Si₃N₄ and SiC etch rate in *c*-C₄F₈ and *c*-C₄F₈/Ar discharges [4]. The results showed that a change in the steady-state fluorocarbon film thickness caused by the addition of N₂ to the *c*-C₄F₈/Ar gas mixture has a negligible effect on Si₃N₄. Takahashi et al. conducted HfO₂ etching experiments in ICP discharge in which Ar was mixed with CF₄ or *c*-C₄F₈, and they found that the etch rate depends on various external and plasma parameters [6].

In particular, Rauf and Balakrishna conducted two-dimensional simulations in a fluorocarbon mixture capacitively coupled plasma (CCP) discharge and calculated the oxide etch rate on various operation conditions [7]. Moreover, Huang et al. conducted both reactor and feature profile simulations in Ar/C₄F₈/O₂ CCP discharges [8]. Although many reactions are considered in simulations, several reaction rate coefficients are still assumed, and some electron collision reactions were neglected. Above all, the collision cross-sections

are essential input data for plasma simulations, and accurate data feeds directly into the accuracy of simulations. Electron impact ionization, dissociation, attachment cross-sections, and momentum transfer and excitation collision cross-sections are essential to obtain more accurate simulation results [9]. Due to the importance of fluorocarbons in plasma applications, there has been a surge of studies for fluorocarbons electron collision cross-sections. Recently, we investigated electron collision studies with $C_2F_2$ [10], $C_3F_4$ [11], and $C_4F_6$ [12] isomers, for which there were very little data in the literature, highlighting the importance of the need for more investigation. The importance of investigating fluorocarbons and other feedstock gases for replacing currently used higher global-warming-potential gases has been highlighted in a recent review [13]. For $c$-$C_4F_8$, experimental data of total and ionization cross-sections are available, but clearly, there is a lack of detailed theoretical investigation for this important target.

Christophorou and Olthooff [14] gave recommended electron-collision data and transport-coefficient data for $c$-$C_4F_8$. Jelisavcic et al. [15] measured the absolute cross-sections for elastic scattering of electrons from $c$-$C_4F_8$ in the energy range of 1.5–100 eV and over the scattering angles of $10°$–$130°$. The most recent recommended data for $c$-$C_4F_8$ was provided by Yoon et al. [16] for the integral elastic ($Q_{el}$), momentum transfer (MTCS), and differential (DCS) cross-sections. Measurements for total cross-sections (TCS) were made for this gas by Makochekanwa et al. [17], Sanabia et al. [18], and Nishimura et al. [19]. Winstead and Mckoy [20] calculated the $Q_{el}$, DCS, and MTCS using the Schwinger multichannel (SMC) using a limited CC and SE approximation. In this article, we concentrate on the low-energy $Q_{el}$, MTCS, DCS, and excitation cross-sections ($Q_{exc}$) using the SE, SEP, and CC approximation using the R-matrix method.

## 2. Theoretical Methodology

The R-matrix method is the most common ab initio method for studying electron-molecule interactions at low energies. Tennyson extensively reviewed and explained the molecular R-matrix method [21,22], and hence, only a brief description of the method will be presented here. The Quantemol Electron Collision (QEC) code [23], which runs both the MOLPRO package [24] and the new version of UK molecular R-matrix code UKRMol+ [25], was used here to study the electron scattering from $c$-$C_4F_8$. QEC is new expert system that replaces Quantemol-N [26] and runs the upgraded R-matrix code. Quantemol-N has been successfully used for low-energy collision cross-section calculations for a variety of molecular targets [27–31]. Initial studies were performed with Quantemol-N, while the final results given below were all obtained using QEC.

In the R-matrix method, we divide the configuration space into an inner and outer region by a sphere of radius ($r = a$). In the inner region, the short-range interactions, such as static, exchange, and polarization, are important, and one needs to consider their effect between the incident projectile and the target under study. The adapted quantum chemistry codes provide the solution of the inner region. The good thing about the inner region problem is that it needs to be solved only once, as they are independent of the energy of the scattering electron.

The wave function for the ($N + 1$) electron system in the CC approximation [32] for the inner region is given as

$$\psi_k^{N+1} = A \sum_{ij} a_{ijk} \Phi_i^N(x_1, \ldots, x_N) u_{ij}(x_{N+1}) \\ + \sum_i b_{ik} \chi_i^{N+1}(x_1, \ldots, x_{N+1})$$

(1)

where $A$ is the anti-symmetrization operator that accounts for the exchange between the target electrons and the scattering electron. The diagonalization of the Hamiltonian in the inner region gives us the variational coefficients $a_{ijk}$ and $b_{ik}$. The scattering electron is represented by the continuum orbitals, $u_{ij}$. In the first term, $\Phi_i^N$ represents the wavefunction of the $i$th target state, and $x_N$ is the spatial and spin coordinate of the $N$th target electron,

and the summation runs over the target plus continuum states used in the close-coupled expansion. The second term contains configurations that represent the short-range correlations and polarization effects called $L^2$ configurations. These are multi-center quadratically integrable functions constructed by placing all the $N + 1$ electrons in the target molecular orbitals (MOs).

In the outer region, the scattering electron moves in the long-range multipole potential of the target molecule, where the effect of dipole and quadrupole moments influences the scattering electron. Hence, in this region, the exchange and correlation effects are minimal, and we consider only the long-range interactions between the projectile and molecular target. In the present calculations, the outer region is extended up to $100a_0$, and the energy dependence of the scattering electron is carried in this region, where all the required quantities, such as eigen phase sum and scattering cross-sections, are calculated.

In the present calculations, Gaussian-type orbitals (GTOs) are used to represent the molecular and continuum orbitals (both occupied and virtual orbitals). The complete molecular orbitals were obtained from the Hartree–Fock Self-Consistent Field (HFSCF) method, and the continuum orbitals used were the GTOs of Faure et al. [33]. After obtaining the solution for the inner region, R-matrix provides the bridge between the inner and the outer region. The R-matrix constructed on the boundary from the inner region solutions is propagated outwards up to $100a_0$ until it is matched with asymptotic functions given by the Gailitis expansion [34]. After matching to the boundary conditions, the symmetric *K*-matrices are determined, and all the observables, such as cross-sections, are obtained using the *K*-matrix elements. Resonances, an essential part of the low-energy calculations, are identified and detected using the RESON [35] module by fitting them to the Briet–Wigner profile [36] to obtain the energies and widths. From the *K*-matrices, we can obtain the *T*-matrices as follows:

$$T = \frac{2ik}{1 - ik} \tag{2}$$

The *T*-matrices in turn were used to calculate various cross-sections. The DCS and MTCS are calculated using the *K*-matrices in the POLYDCS program of Sanna and Gianturco [37].

We used the three different scattering models of SE, SEP, and CC to model the scattering processes. In the SE model, all the target electrons are kept in the ground-state configuration (frozen electrons). In this configuration, HFSCF target wavefunctions are used and are not allowed to be polarized by the incident electron. However, this approximation is well-suited for detecting shape resonances but not good enough for detecting Feshbach/core-excited, which involves the excitation of bound electrons. The logical advance to the SE model is the SEP approximation, where the effect of target polarization is taken into account. The polarization effects are taken into account by promoting an electron from a target to a virtual orbital, and also, the scattering electron is put into the virtual orbital giving two particles and one hole configuration [21]. In the R-matrix method, this is done by the use of $L^2$ configuration in Equation (1). The third model (CC) is a more sophisticated approximation than SE and SEP, and in this case, one can include many electronic excited states into the calculation in the expansion of equation (1). Here, some electrons are frozen in the ground, and some are allowed to move freely in the active space, which helps to incorporate electronic states into calculations, leading to the much better description of the polarization effects, which gives us the more accurate cross-sections and resonance energies. Due to the inclusion of the excited states into the calculations, this approximation is well-suited for detecting Feshbach/core-excited resonances at low energies.

*Target Models*

The Gaussian 09 [38] program suite was used to optimize the geometry of $c$-$C_4F_8$. The equilibrium geometry of $c$-$C_4F_8$ was obtained by fully optimizing the molecular structure and orbital parameters using DFT-$\omega$B97X-D [39] hybrid functionals and Dunning's [40] aug-cc-pVTZ basis set. The optimized structure of $c$-$C_4F_8$ is given in Figure 1.

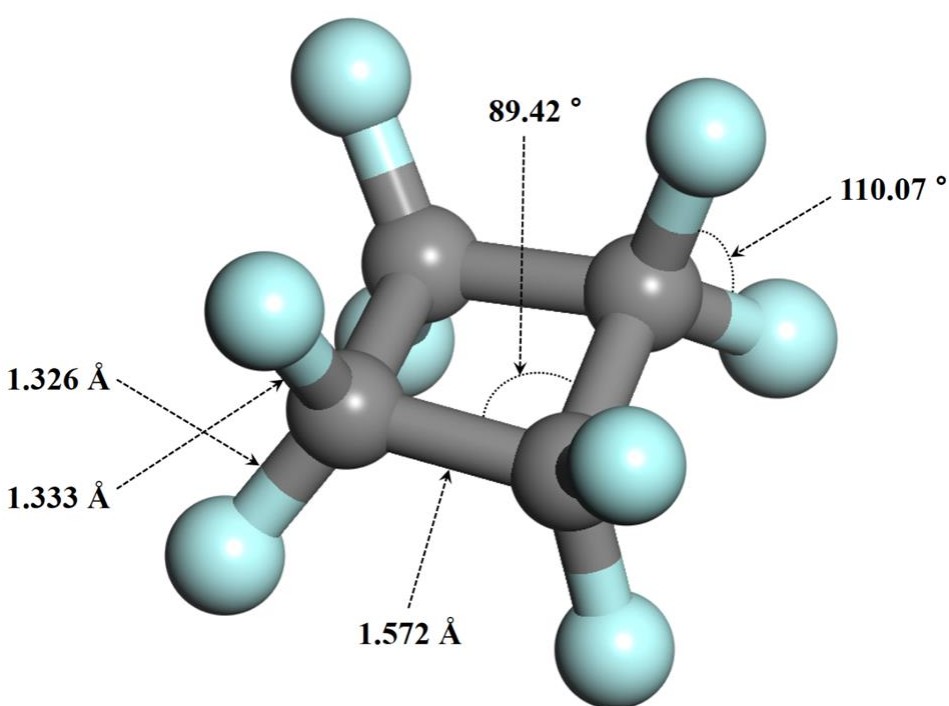

**Figure 1.** Equilibrium structure of neutral $c$-$C_4F_8$ at $\omega$B97X-D/aug-cc-pVTZ. The ring-puckering angle is 16.33°.

$c$-$C_4F_8$ is a closed-shell target that belongs to a $D_{2d}$ point group symmetry. The calculations are performed in the $C_{2v}$ symmetry, which is the subgroup of $D_{2d}$. The ground-state Hartree–Fock electronic configuration of $c$-$C_4F_8$ molecule is $1b_2^2$, $1a_1^2$, $1b_1^2$, $2a_1^2$, $2b_1^2$, $3a_1^2$, $4a_1^2$, $2b_2^2$, $5a_1^2$, $3b_2^2$, $3b_1^2$, $6a_1^2$, $7a_1^2$, $4b_2^2$, $4b_1^2$, $8a_1^2$, $9a_1^2$, $5b_1^2$, $5b_2^2$, $10a_1^2$, $11a_1^2$, $6b_2^2$, $6b_1^2$, $12a_1^2$, $13a_1^2$, $1a_2^2$, $14a_1^2$, $7b_2^2$, $7b_1^2$, $8b_1^2$, $8b_2^2$, $15a_1^2$, $2a_2^2$, $9b_2^2$, $9b_1^2$, $16a_1^2$, $3a_2^2$, $10b_2^2$, $10b_1^2$, $17a_1^2$, $4a_2^2$, $18a_1^2$, $11b_2^2$, $11b_1^2$, $19a_1^2$, $12b_2^2$, $12b_1^2$, $5a_2^2$ in the $C_{2v}$ symmetry. For the scattering calculations, we used the complete active space–configuration interaction (CAS-CI) model to represent the target wavefunction with cc-pVTZ basis set and 15 target states in our calculations. Of 96 electrons, only 8 electrons are located in the active space composed of $19a_1$, $20a_1$, $21a_1$, $12b_1$, $13b_1$, $12b_2$, $13b_2$, and $5a_2$ molecular orbitals in this CAS-CI model. The accuracy of the scattering data depends on the choice of the target wave function, and hence, careful and critical assessment of the target wavefunction is essential. The number of configuration-state functions (CSF) generated for the ground state is 900, and 225 channels are included in the present scattering calculation. To accommodate the target electrons' charge cloud inside the inner region, the inner region radius of $10a_0$ was sufficient and provided a stable calculation in the present case. Two virtual orbitals were included in the SE, SEP, and CC scattering calculations.

The present CC model predicts the ground-state energy of $c$-$C_4F_8$ to be $-946.85$ Hartree. The triplet and singlet excited-states thresholds are 8.80 and 9.04 eV, which compares well with the 8.52 and 9.13 eV triplet and singlet excited-states thresholds computed using the single-excitation configuration-interactions (SECI) calculations [20]. At the energy minimum of the ground state, the vertical excitation energies to the six lowest-lying electronic excited singlets and triplets are provided in Table 1.

**Table 1.** Electronic vertical excitation energy at the $c$-$C_4F_8$ ground-state geometry. The results are compared with previous theoretical calculations [20] in eV.

| State ($C_{2v}$) | Present | Theory [20] |
|:---:|:---:|:---:|
| $^1A_1$ | 0 | |
| $^3A_2$ | 8.80 | 8.52 |
| $^1A_2$ | 9.04 | 9.13 |
| $^3A_2$ | 11.54 | |
| $^3E$ | 12.23 | |
| $^1A_2$ | 12.45 | |
| $^1E$ | 13.53 | |

## 3. Results and Discussion

Figure 2 shows the eigenphase diagram for various doublet scattering states $^2A_1$, $^2E$, and $^2A_2$ of the $c$-$C_4F_8$ system using the SE, SEP, and CC models in the $C_{2v}$ point group symmetry. The eigenphase diagram is important for the study of resonances at low-energy regimes. In Figure 2, the scattering state $^2A_1$ shows a hump at around 3~4 eV with all three models. As expected, the SE model detected the resonance at slightly higher energy than the other two models due to the exclusion of the polarization and correlation effects in its calculations. The SEP model detected a shape resonance at 3.12 eV and a Feshbach resonance at 7.73 eV, as indicated by a hump at the same energies in the eigenphase sum due to the $^2A_1$ and $^2E$ scattering states. The position of the resonance and their corresponding widths for $c$-$C_4F_8$ below 10 eV are presented in Table 2 along with the resonance data of Winstead and Mckoy [20] and the experimental dissociative electron attachment (DEA) thresholds [41–44], which can be associated with resonance at different positions.

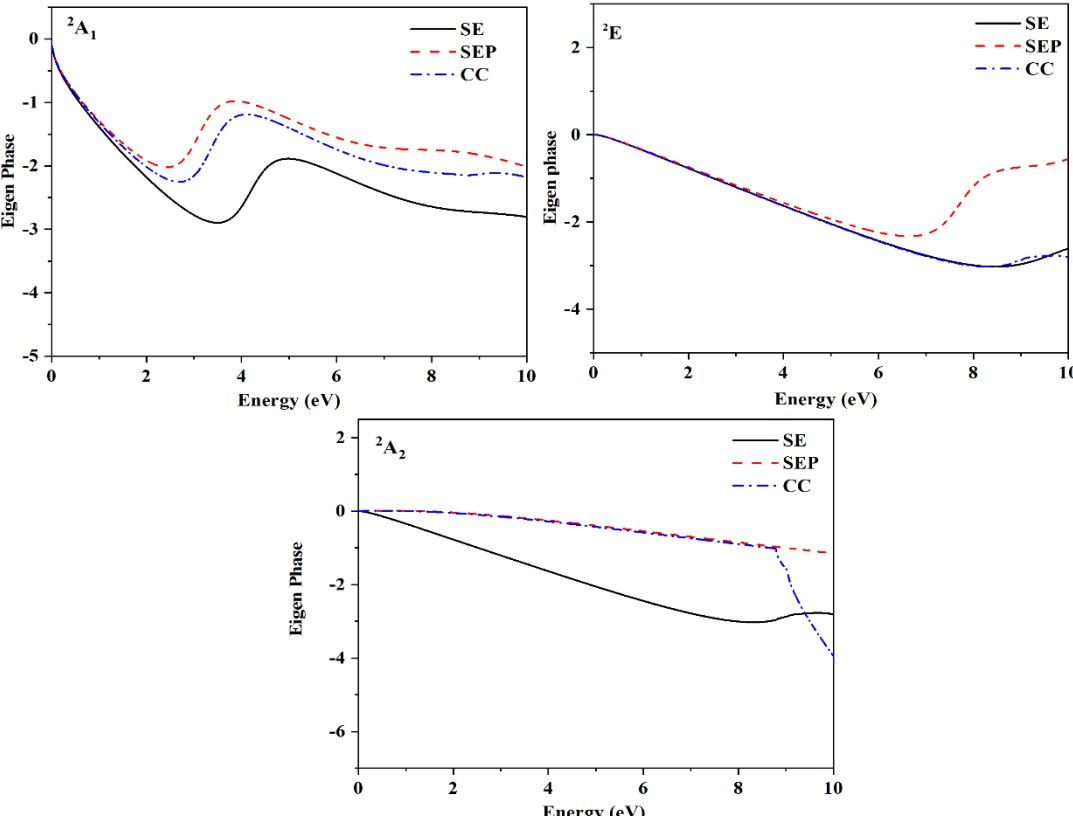

**Figure 2.** Eigen phase diagram for the doublet scattering states of e-$c$-$C_4F_8$ scattering in $C_{2v}$ symmetry.

**Table 2.** Resonance position and widths for $c$-$C_4F_8$ were detected in the present calculations in eV along with the comparison of Winstead and Mckoy [20] and experimental DEA [41–44] data.

| States ($C_{2v}$) | Present (SE) | | Present (SEP) | | Present (CC) | | Winstead and Mckoy [20] | | Experimental DEA Results |
| --- | --- | --- | --- | --- | --- | --- | --- | --- | --- |
| | Position | Width | Position | Width | Position | Width | Position | Width | Position |
| $^2A_1$ | 4.21 | 1.01 | 3.12 | 0.95 | 3.37 | 0.92 | 3.0 | 0.33 | 3.75 [41] |
| $^2E$ | | | 7.73 | 1.07 | | | 8.1 | 1.2 | 8.0 [41], 8.2 [42], 8.5~8.8 [43], 7.9 [44] |

Figure 3 shows the $Q_{el}$ and MTCS from the R-matrix method's SE, SEP, and CC models. The results obtained with different models are compared with the experimental, theoretical, and recommended datasets. In Figure 3a,c, the $Q_{el}$ and MTCS are compared that are calculated using different approximations. All the models detect the presence of a shape resonance at around 3~4 eV, with the SEP model predicting the resonance at lower energy compared to the other two models, as seen for both $Q_{el}$ and MTCS. The 7.73 eV resonance is detected in the SEP model and is also supported by the eigenphase diagram. The shape resonance is detected at 4.21, 3.12, and 3.37 eV, with the corresponding widths of 1.01, 0.95, and 0.92 eV in the SE, SEP, and CC models, respectively.

Here, we discuss only the resonances detected below 10 eV in our calculations, which are compared with those of the calculations of Winstead and Mckoy [20] and the experimental data for the dissociative electron attachment. The shape resonance detected with all the three models in the present calculation at around 3~4 eV due to $^2A_1$ state of the $C_{2V}$ symmetry is associated with the $^2B_2$ resonance of Winsted and Mckoy [20] in the $D_{2d}$ symmetry. The Feshbach resonance detected at 7.73 eV due to $^2E$ state with the SEP model in the present calculations can be associated with the 8.1 eV resonance of Winstead and Mckoy due to the $^2E$ state in the $D_{2d}$ symmetry. The present shape resonance at 3.12, 3.37, and 4.21 eV due to SEP, CC, and SE models could also be associated with the 3.75 eV observed dissociative attachment maximum to $c$-$C_4F_8$ of Lifshitz and Grajower [41]. The present Feshbach resonance at 7.73 eV could be associated with the experimental dissociative electron attachment to $c$-$C_4F_8$ at 8.0, 8.2, 8.5~8.8, and 7.9 eV due to Lifshitz and Grajower [41], Bibby and Carter [42], Harland and Thynne [43], and Sauers et al. [44].

The present $Q_{el}$ and MTCS results are compared in Figure 3b,d with other available datasets for elastic and total cross-sections. The present and the data of Winstead and Mckoy [20] for $Q_{el}$ show a large disagreement with the experimental data [18,19] and recommended dataset of Christophorou [14] and Yoon et al. [16] at low energies below 8 eV, after which they follow the experimental and recommended data. The shape resonance detected in the previous and the present calculations is missing in the experimental data [18,19] for total cross-section. Since the target is quite large and complex, the effects of correlation and polarization are not sufficiently well-modelled, which may be a cause of the discrepancy between the experiment and the theoretical calculations at low energies. The MTCS follows the calculation of Winstead and Mckoy [20] but is in less good agreement with the recommended data of Yoon et al. [16].

Figure 4 shows the pictorial representation of the elastic DCS for various energies. The DCS is plotted for energies of 1.5, 3, 5, 7, 8, and 10 eV and are compared with the previous results of Winstead and Mckoy [20] for energies of 1.5, 5, 8, and 10 eV and with the recent recommended dataset of Yoon et al. [16] for 1.5, 3, 5, 7, 8, and 10 eV. The DCS shows a maximum value at the forward scattering angle, and it decreases slowly as the angle of scattering is increased. The DCS minima occurs at around 110°–120° for 5, 7, 8, and 10 eV energies,, and slowly it rises again at the backward angles. The present SE, SEP, and CC calculations compared quite well in general with the calculations of Winstead and Mckoy for all the energies. For lower energies until 5 eV, the present DCS does not agree with the recommended data of Yoon et al. [16], but it shows an improvement as the energy increases beyond 5 eV for 7, 8, and 10 eV. Below 7 eV, short-range correlation and polarization effects

play an important role in the target–projectile interactions. Since the target in the present case is quite big, the present approximation did not include sufficient polarization effects at low energies, and hence, that may be one of the reasons for the disagreement with the experiment.

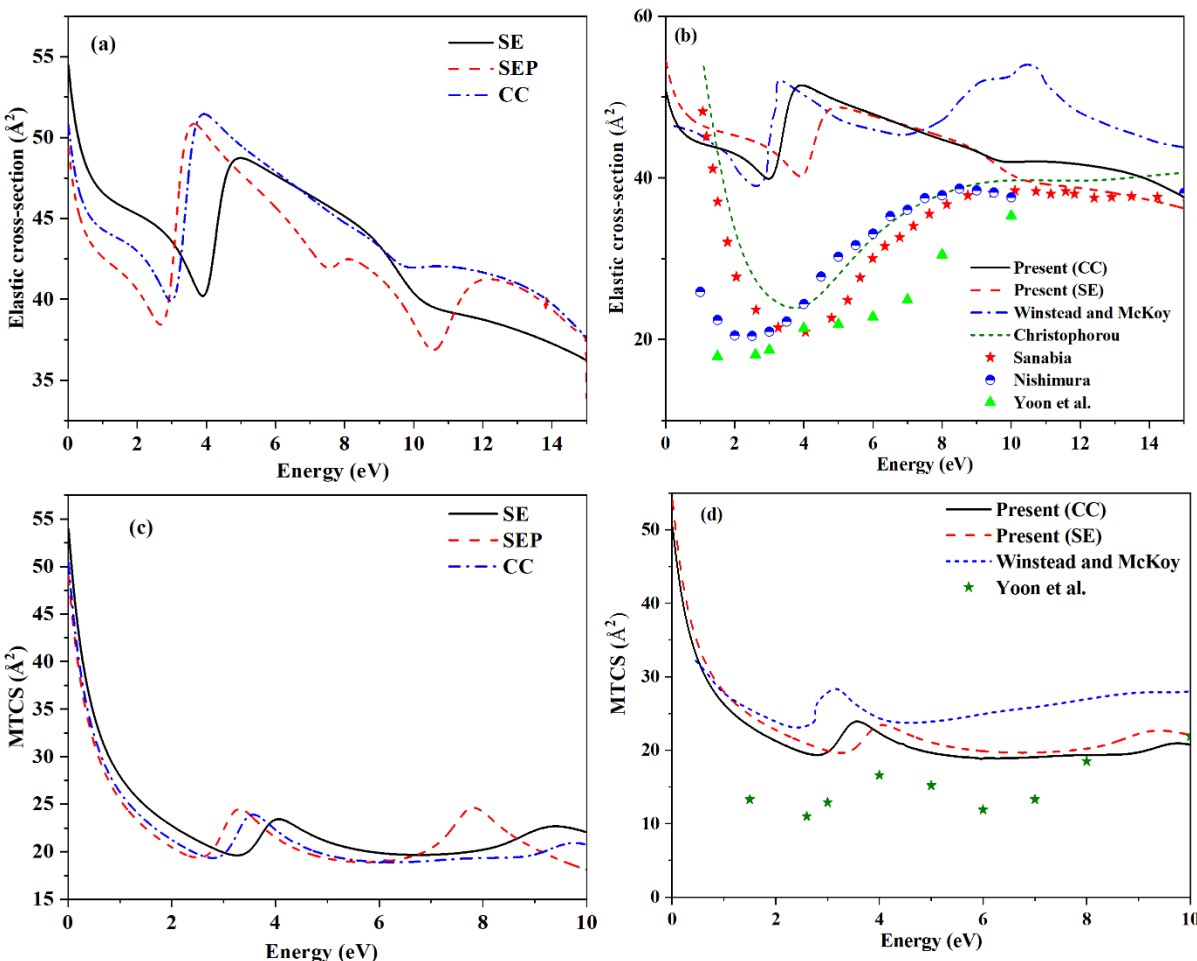

**Figure 3.** Elastic and momentum transfer cross-section of $c$-C$_4$F$_8$ scattering; (**a**) comparison among the different models (SE, SEP, and CC) for the elastic cross-section; (**b**) comparison of the present elastic cross-section (SE and CC) with the available data in the literature; (**c**) comparison among the different models (SE, SEP, and CC) for the MTCS; (**d**) comparison of the present MTCS (SE and CC) with the available data in the literature.

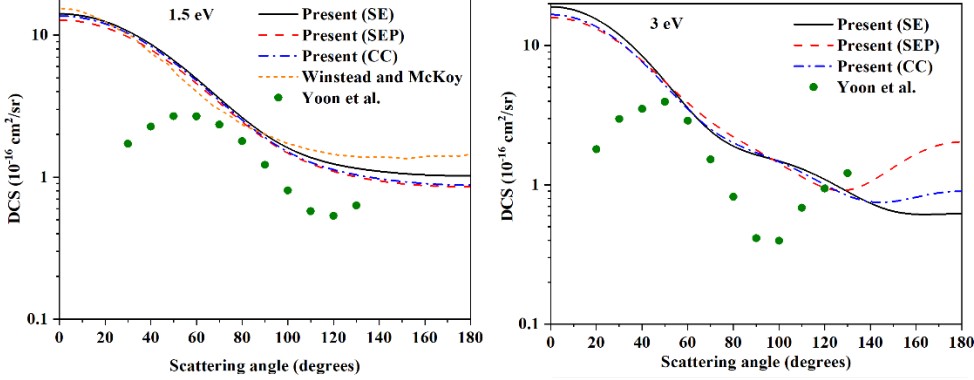

**Figure 4.** *Cont.*

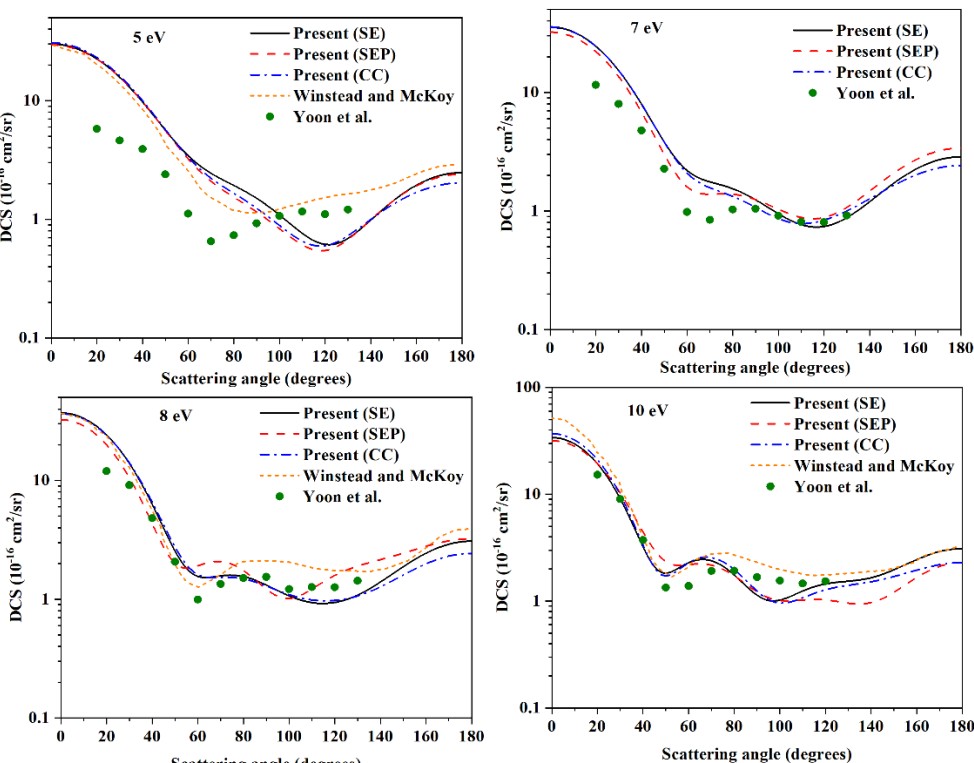

**Figure 4.** Elastic DCS for the electron scattering of *c*-C$_4$F$_8$ system for the energies of 1.5, 3, 5, 7, 8, and 10 eV.

Figure 5 depicts the electronic $Q_{exc}$ from the ground state of *c*-C$_4$F$_8$ to the six low-lying excited states. The vertical excitation threshold of the first excited state ($^3$A$_2$) is around 8.8 eV. The triplet states contributes maximum to the $Q_{exc}$, and for $^3$E excited state, a maximum cross-section is found approximately at 15 eV.

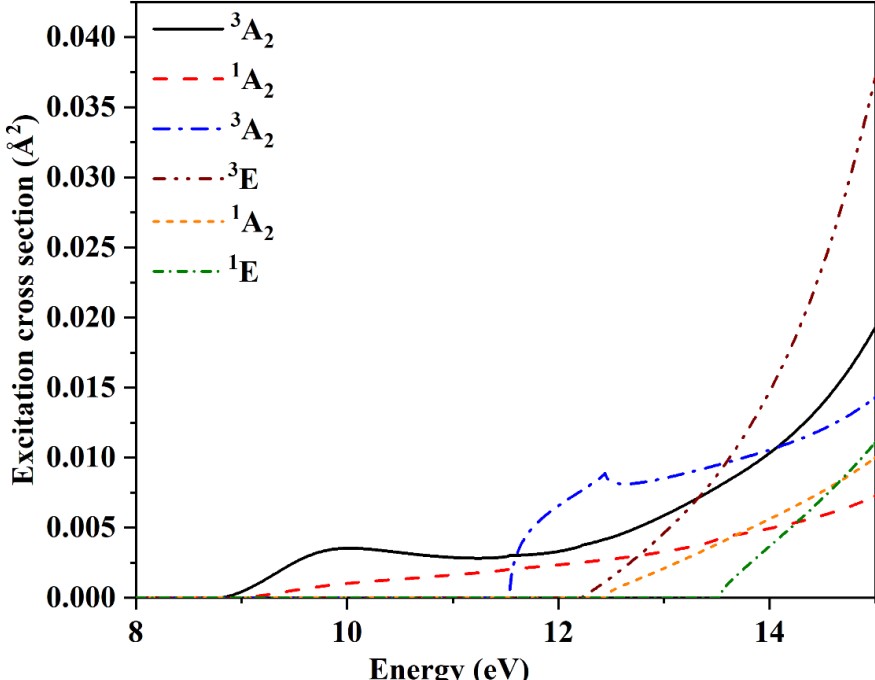

**Figure 5.** Electronic excitation cross-section from the ground state to the six low-lying excited states of *c*-C$_4$F$_8$ scattering.

**Summary:** This work investigates electron collision study of plasma-relevant molecular $c$-$C_4F_8$ target using the SE, SEP, and CC models. The present calculations have reproduced the previous theoretical results [20] calculated using a similar approximation. We could also confirm the presence of shape resonance at around 3~4 eV and Feshbach resonance at 7.73 eV, in accordance with the earlier calculation and the experimental dissociative attachment study to $c$-$C_4F_8$. The present study suggests that we could use similar models and approximations to study more complex targets, such as $c$-$C_5F_8$, $c$-$C_6F_8$, $C_7F_8$, $C_7F_{14}$, and $c$-$C_{10}F_8$, and test their validity for replacing the PFC gases with higher global-warming-potential ones, as highlighted in a recent review article [13]. It is quite clear that there is lack of studies for larger fluorocarbons, and even for a smaller targets, scarcity of data is seen, and hence, we hope this study can motivate others to investigate more on this subject. Moreover, the present data would find applications in low-temperature plasma modelling and simulation.

**Author Contributions:** Conceptualization, M.-Y.S., D.G. and H.C.; Methodology, D.G., J.T. and H.S.; data curation, D.G., H.S. and H.C.; original draft preparation, D.G., D.-C.K. and H.C, review and editing, J.T., J.-S.Y. and M.-Y.S. All authors have read and agreed to the published version of the manuscript.

**Funding:** This research received no external funding.

**Data Availability Statement:** The data relevant to the study is available with authors upon reasonable request.

**Acknowledgments:** D.G. is pleased to acknowledge Vellore Institute of Technology, Vellore, for support. H.C. and M.S. acknowledges support from the R + D Program Plasma BigData ICT Convergence Technology Research Project through the Korea Institute of Fusion Energy (KFE), funded by the Government, Republic of Korea. He Su acknowledges support from the Chinese Scholarship Council and the support from National Key R&D Program of China (Grant 2017YFA03036000) for her visit to UCL. The authors also thank the Korea Institute of Energy Research, South Korea, and University College London for providing the resources needed for calculations.

**Conflicts of Interest:** The authors declare no conflict of interest.

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
