# Peer review of "Low-Energy Electron Scattering from c-C4F8"

_atoms, doi:10.3390/atoms10020063_

Round 1

Reviewer 1 Report

This manuscript reports new calculations of electron scattering from c-C4F8 using the R-matrix method. These computations supplement the existing Schwinger multichannel method calculations and some experimental studies. The present results seem consistent with the available previous theoretical calculations. The present manuscript is suitable for publication after the following details are addressed.

(1)         There is large difference between the experimental and theoretical elastic DCS and ICS at low energies. Perhaps a comment about the reliability of the theory is in order to suggest if theoretical approximations are breaking down at these low energies?

The present manuscript could also use some typesetting/formatting degree symbol should be “°” and not “0” and some punctuation (Periods used in incorrect locations and used instead of commas) and English language corrected.

Reference 19 seems incorrect in the reference list: should it be

Journal of the Physical Society of Japan, January 15, 2007, Vol. 76, No. 1

Total Electron Scattering Cross Sections for Cyclofluorobutane

Hiroyuki NishimuraAkira Hamada

https://doi.org/10.1143/JPSJ.76.014301

Reviewer 2 Report

This manuscript presents a theoretical study of low-energy electron scattering from the molecule c-C4F8.  As an experimentalist, I cannot comment on the theoretical approaches used; but as a "general reader" I found the information to be well presented.  In addition, the introductions supplies a nice, comprehensive background of the topic.
However, after reading the manuscript I am left with the question "what have we learned?"  The summary contains phrases such as "the present calculatons have reproduced ...", "we could also confirm ...." and "the present data could be helpful ...." without any indications as to whether previous calculations are in doubt or exactly how the present study is helpful in either answering an important question or as a stepping stone to understanding other, or more difficult to model, systems. As presented, the impression I am left with is "we did the following study because we could and here are the results.   Also, considering the theoretical-experimental comparisons shown in Figures 3b and 3d and for energies below 7eV in Figure 4, I find the statement about "good agreement with the recommended dataset" to be controversial. My recommendation is that these points need to be addressing and a modified manuscript be submitted for further review.

Round 2

Reviewer 2 Report

I thank the authors for the modifications they have made as suggested in my previous review.  I now find the revised manuscript suitable for publication and hope to see some future work based on the present study where improved models for correlation and polarization are added in order to improve the results at low energies.